# LEAP: LEARNING EMBEDDINGS FOR ADAPTIVE PACE

## ABSTRACT

Determining the optimal order in which data examples are presented to Deep Neural Networks during training is a non-trivial problem. However, choosing a non-trivial scheduling method may drastically improve convergence. In this paper, we propose a Self-Paced Learning (SPL)-fused Deep Metric Learning (DML) framework, which we call Learning Embeddings for Adaptive Pace (LEAP). Our method parameterizes mini-batches dynamically based on the *easiness* and *true diverseness* of the sample within a salient feature representation space. In LEAP, we train an *embedding* Convolutional Neural Network (CNN) to learn an expressive representation space by adaptive density discrimination using the Magnet Loss. The *student* CNN classifier dynamically selects samples to form a mini-batch based on the *easiness* from cross-entropy losses and *true diverseness* of examples from the representation space sculpted by the *embedding* CNN. We evaluate LEAP using deep CNN architectures for the task of supervised image classification on MNIST, FashionMNIST, CIFAR-10, CIFAR-100, and SVHN. We show that the LEAP framework converges faster with respect to the number of mini-batch updates required to achieve a comparable or better test performance on each of the datasets.

## 1 INTRODUCTION

The standard method to train Deep Neural Networks (DNNs) is stochastic gradient descent (SGD) which employs backpropagation to compute gradients. It typically relies on fixed-size mini-batches of random samples drawn from a finite dataset. However, the contribution of each sample during model training varies across training iterations and configurations of the model's parameters (Lapedriza et al., 2013). This raises the importance of *data scheduling* for training DNNs, that is, searching for an optimal ordering of training examples which are presented to the model. Previous studies on Curriculum Learning (Bengio et al., 2009, CL) show that organizing training samples based on the ascending order of difficulty can favour model training. However, in CL, the curriculum remains fixed over the iterations and is determined without any knowledge or introspection of the model's learning. Self-Paced Learning (Kumar et al., 2010) presents a method for dynamically generating a curriculum by biasing samples based on their *easiness* under the current model parameters. This can lead to a highly imbalanced selection of samples, i.e. very few instances of some classes are chosen, which negatively affects the training process due to overfitting. Loshchilov & Hutter (2015) propose a simple batch selection strategy based on the loss values of training data for speeding up neural network training. However, their results are limited and the approach is time-consuming, as it achieves high performance on MNIST, but fails on CIFAR-10. Their work reveals that selecting the examples to present to a DNN is non-trivial, yet the strategy of uniformly sampling the training data set is not necessarily the optimal choice.

Jiang et al. (2014b) show that partitioning the data into groups with respect to *diversity* and *easiness* in their Self-Paced Learning with Diversity (SPLD) framework, can have substantial effect on training. Rather than constraining the model to limited groups and areas, they propose to spread the sample selection as wide as possible to obtain diverse samples of similar easiness. However, their use of $K$-Means and Spectral Clustering to partition the data into groups can lead to sub-optimal clustering results when learning non-linear feature representations. Therefore, learning an appropriate metric by which to capture similarity among arbitrary groups of data is of great practical importance. Deep Metric Learning (DML) approaches have recently attracted considerable attention and have been the focus of numerous studies (Bell & Bala (2015); Schroff et al. (2015)). The

most common methods are supervised, in which a feature space in which distance corresponds to class similarity is obtained. The Magnet Loss (Rippel et al., 2015) presents state-of-the-art performance on fine-grained classification tasks. Song et al. (2016a) show that it achieves state-of-the-art on clustering and retrieval tasks.

This paper makes two key contributions toward scheduling data examples in the mini-batch setting:

- We propose a general sample selection framework called Learning Embeddings for Adaptive Pace (LEAP) that is independent of model architecture or objective, and learns when to introduce certain samples to the DNN during training.

- To our knowledge, we are the first to leverage metric learning to improve self-paced learning. We exploit a new type of knowledge —similar instance-level samples are discovered through an *embedding* network trained by DML in concert with the self-paced learner.

## 2 RELEVANT WORK

### 2.1 LEARNING SMALL AND EASY

The perspective of "starting small and easy" for structuring the learning regimen of neural networks dates back decades to Elman (1993). Recent studies show that selecting a subset of *good* samples for training a classifier can lead to better results than using all the samples (Lee & Grauman, 2011; Lapedriza et al., 2013). Pioneering work in this direction is Curriculum Learning (Bengio et al., 2009), which introduced a heuristic measure of easiness to determine the selection of samples from the training data. By comparison, SPL (Kumar et al., 2010) quantifies the easiness by the current sample loss. The training instances with loss values larger than a threshold, $\lambda$, are neglected during training and $\lambda$ dynamically increases in the training process to include more complex samples, until all training instances are considered. This theory has been widely applied to various problems, including dictionary learning for image classification (Tang et al., 2012), object detection (Sangineto et al., 2016), multimedia event detection (Jiang et al., 2014a), long-term tracking (Supancic III & Ramanan, 2013), visual tracking (Huang et al., 2017) and medical imaging analysis (Li et al., 2017). In SPLD (Jiang et al., 2014b), training data are pre-clustered in order to balance the selection of the easiest samples with a sufficient inter-cluster diversity. However, the clusters and the feature space are fixed: they do not depend on the current self-paced training iteration. Adaptation of this method to a deep-learning scenario, where the feature space changes during learning, is non-trivial. Our self-paced sample selection framework aims at a similar goal but the diversity of samples is obtained with a DML approach to adaptively sculpt a representation space by autonomously identifying and respecting intra-class variation and inter-class similarity.

### 2.2 LEARNING REPRESENTATIONS BY METRIC EMBEDDING

Deep metric learning has gained much popularity in recent years, along with the success of deep learning. The objective of DML is to learn a distance metric consistent with a given set of constraints, which usually aim to minimize the distances between pairs of data points from the same class and maximize the distances between pairs of data points from different classes. DML approaches have shown promising results on various tasks, such as semantic segmentation (Fathi et al., 2017), visual product search (Kiapour et al., 2015), face recognition (Schroff et al., 2015), feature matching (Choy et al., 2016), fine-grained image classification (Zhang et al., 2015), zero-shot learning (Frome et al., 2013) and collaborative filtering (Hsieh et al., 2017). DML can also be used for challenging, extreme classification settings, where the number of classes is very large and the number of examples per class becomes scarce. Most of the current methods define the loss in terms of pairs (Song et al., 2016b; Ustinova & Lempitsky, 2016), triplets (Schroff et al., 2015; Wang et al., 2017) or $n$-pair tuples (Sohn, 2016) inside the training mini-batch. These methods require a separate data preparation stage which has very expensive time and space cost. Also, they do not take the global structure of the embedding space into consideration, which can result in reduced clustering. An alternative is the Magnet Loss (Rippel et al., 2015) and DML via Facility Location (Song et al., 2016a) which do not require the training data to be preprocessed in rigid paired format and are aware of the global structure of the embedding space. Our work employs the Magnet loss to learn a representation space, where we compute centroids on the raw features and then update the learned representation continuously.

To our knowledge, the concept of employing DML for SPL-based DNN training has not been investigated. Effectively, an end-to-end DML can be constructed to be a feature extractor using a deep CNN which can learn to sculpt an expressive representation space by metric embedding. We can use this feature representation space which maintains intra-class variations and inter-class similarity to select samples based on the *true diverseness* and *easiness* for the student model we want to train. Our architecture combines the strength of adaptive sampling with that of mini-batch online learning and adaptive representation learning to formulate a representative self-paced strategy in an end-to-end DNN training protocol.

## 3 LEARNING EMBEDDINGS FOR ADAPTIVE PACE (LEAP)

The Learning Embeddings for Adaptive Pace (LEAP) framework consists of a dual DNN setup. An *embedding* DNN learns a salient representation space, then transfers its knowledge to the self-paced selection strategy to train the second DNN, called the *student*. In this work, but without loss of generality, we focus on training deep Convolutional Neural Networks (CNNs) for the task of supervised image classification. More specifically, an *embedding* CNN is trained alongside the *student* CNN of ultimate interest (Figure 1). In this framework, we want to form mini-batches using the *easiness* and *true diverseness* as sample importance priors for the selection of training samples. Given that we are learning the representation space adaptively alongside the student as training progresses, this has negligible computational cost compared to the actual training of the student CNN (see Section 4).

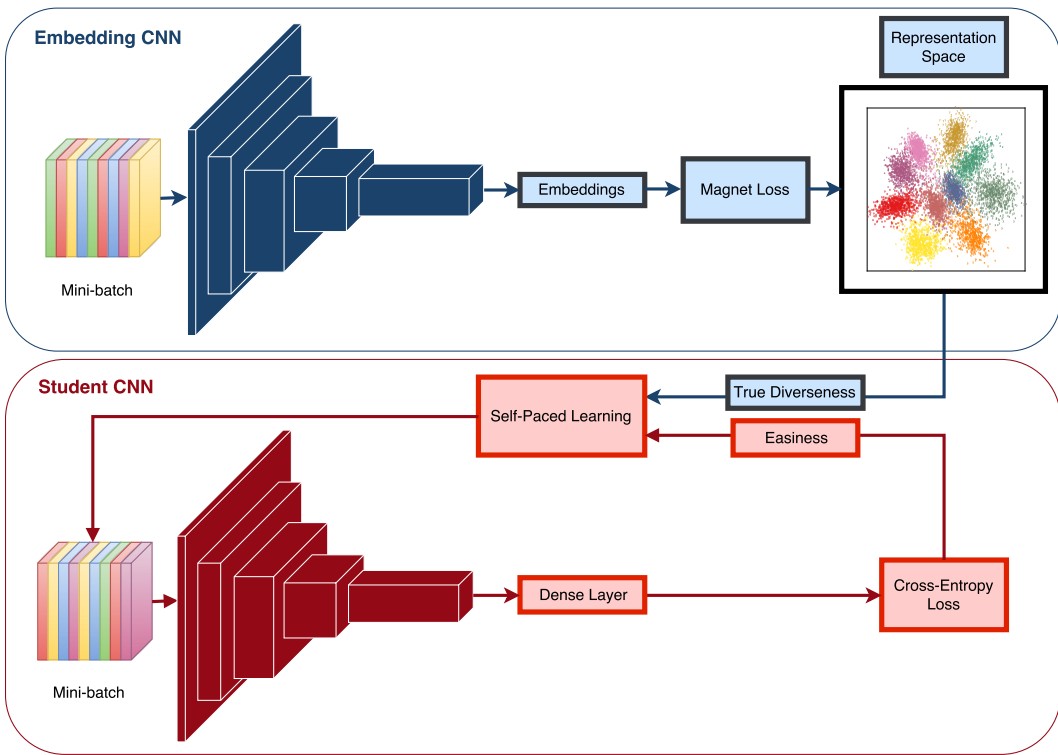

FIGURE 1: The LEAP framework, consisting of an embedding CNN that learns a representation for the student CNN to create a self-paced strategy based on *easiness* and *true diverseness* as sample importance priors.

### 3.1 EMBEDDING CNN

We adopt the Magnet loss to learn a representation space because it proved to achieve a higher accuracy on classification tasks, in comparison to margin-based Triplet loss and softmax regression (Rippel et al., 2015). Assuming we have a training set with $N$ input-label pairs $\mathcal{D} = \{x_n, y_n\}_{n=1}^{N}$,

the Magnet loss learns the distribution of distances for each example, from $K$ clusters assigned for each class, $c$, denoted as $\{\mathcal{I}_k^c\}_{k=1}^K \ \forall \ C$ classes. The mapping of inputs to representation space are parametrized by $\mathbf{f}(\cdot; \Theta)$, where their representations are defined as $\mathbf{r}_n = \{\mathbf{f}(x_n; \Theta)\}_{n=1}^N$. The approach then repositions the different cluster assignments using an intermediate $K$-Means++ clustering (Arthur & Vassilvitskii, 2007). Therefore, for each class $c$, we have, $\{\mathcal{I}_k^c\}_{k=1}^K = \underset{I_1^c, \ldots, I_K^c}{\operatorname{argmin}} \sum_{k=1}^K \sum_{r \in I_k^c} \|\mathbf{r} - \mu_k^c\|_2^2$ and $\mu_k^c = \frac{1}{\|I_k^c\|} \sum_{\mathbf{r} \in I_k^c} \mathbf{r}$.

The class of representation $\mathbf{r}$ and its assigned cluster center are defined as $C(\mathbf{r})$ and $\mu(\mathbf{r})$, respectively. The mini-batches used to train the embedding CNN are constructed iteratively with neighbourhood sampling. Moreover, a seed cluster is sampled $I_1 \sim p_{\mathcal{I}}(\cdot)$, then the nearest $M - 1$ impostor clusters is retrieved from $\{I_m\}_{m=2}^M$ of $I_1$. Finally, $D$ training instances $\{x_d^m\}_{d=1}^D \sim p_{I_m}(\cdot)$ are uniformly sampled for each cluster $\{I_m\}_{m=1}^M$. The losses of each example is stored and the average loss $\mathcal{L}_I$ of each cluster $I$ is computed during training. This results in the following stochastic approximation of the Magnet Loss objective:

$$\hat{\mathcal{L}}(\Theta) = \frac{1}{MD} \sum_{m=1}^M \sum_{d=1}^D \left\{ -\log \frac{e^{-\frac{1}{2\hat{\sigma}^2} \|r_d^m - \hat{\mu}_m\|_2^2 - \alpha}}{\sum_{\hat{\mu}: C(\hat{\mu}) \neq C(r_d^m)} e^{-\frac{1}{2\hat{\sigma}^2} \|r_d^m - \hat{\mu}\|_2^2}} \right\}_+, \tag{1}$$

where $\{\cdot\}_+$ is the hinge function, $\alpha \in \mathbb{R}$ is a scalar, the cluster means approximation is $\hat{\mu}_m = \frac{1}{D} \sum_{d=1}^D \mathbf{r}_d^m$, and the variance of all samples from their respective centers is given by $\hat{\sigma} = \frac{1}{MD-1} \sum_{m-1}^M \sum_{d=1}^D \|r_d^m - \hat{\mu}_m\|_2^2$. The process for training the embedding CNN is listed in Algorithm 1.

---

**Algorithm 1:** Train Embedding CNN with Magnet Loss.

**Input** : $\mathcal{D}$ (training data), $B$ (mini-batch size), $E$ (epochs), $R$ (cluster refresh interval), $C$ (number of classes)

**Output:** A representation space consisting of $K$ clusters for $C$ classes: $\mathcal{D}_1^c, \ldots, \mathcal{D}_K^C$.

1 Initialize the embedding CNN model.
2 Initialize $N$. /* Total training samples in $\mathcal{D}$ */
3 Shuffle $\mathcal{D}$ and get the sequence of mini-batches $\mathcal{D} = \left\{\{x_B\}_1, \ldots, \{x_B\}_{\frac{N}{B}}\right\}$.
4 **for** *each mini-batch* $\left\{\{x_B\}_{\frac{N}{B}} = 1, \ldots, \frac{N}{B}\right\}$ **do**
5     /* Get initial embeddings from the input training samples. */
6     Extract features from an intermediate layer of the embedding CNN model.
7 **end**
8 **for** *each class* $\{c = 1, \ldots, C\}$ **do**
9     /* Recompute clusters and store cluster assignments using the extracted features of the whole training set. */
10     Initialize each index with K-means++ to maintain a K-means index for each class, $c$.
11 **end**
12 Perform neighbourhood sampling to construct a mini-batch $x_D$ with $D$ samples, where $D = B$.
13 /* Train Embedding CNN. */
14 **for** *each epoch* $\{e = 1, \ldots, E\}$ **do**
15     Predict using the embedding CNN for $x_D$.
16     Compute the Magnet loss (Eq. 1) using the outputs from the embedding CNN in **step** 14.
17     Cache the new losses.
18     Update the respective cluster losses based on the losses associated with the each sample.
19     /* Refresh representation index */
20     **if** $e == R$ **then**
21        Recompute the representations from the training samples using **steps** $4 - 7$.
22        Update cluster assignments using **steps** $8 - 11$.
23     **end**
24     Sample the next training batch $x_D$ using **step** 12.
25 **end**
26 **return** $\mathcal{D}_1^c, \ldots, \mathcal{D}_K^C$

---

### 3.2 LEAP FRAMEWORK

The aim of LEAP can be formally described as follows. Let us assume that a training set $\mathcal{D}$ consisting of $N$ samples, $\mathcal{D} = \{\mathbf{x}_n\}_{n=1}^N$ is grouped into $K$ clusters for $C$ classes using Algorithm 1. Therefore, we have $\{\mathcal{D}^k\}_{k=1}^K$, where $\mathcal{D}^k$ corresponds to the $k^{th}$ cluster, $n_k$ is the number of samples in each cluster and $\sum_{k=1}^K n_k = N$. The weight vector is denoted accordingly as $\{\mathcal{W}^k\}_{k=1}^K$, where $\mathcal{W}^k = (\mathcal{W}_1^k, \ldots, \mathcal{W}_{n_k}^k)^T \in [0,1]^{n_k}$. Non-zero weights of $\mathcal{W}$ are assigned to samples that the student model considers "easy" and non-zero elements are distributed across more clusters to increase diversity. This leads to an objective similar to the one presented in SPLD:

$$\min_{\theta, \mathcal{W}} \mathbb{E}(\theta, \mathcal{W}; \lambda, \gamma) = \sum_{i=1}^N w_i \mathcal{L}(y_i, f(x_i, \theta)) - \lambda \sum_{i=1}^N w_i - \gamma \|\mathcal{W}\|_{2,1}, \text{ s.t } \mathcal{W} \in [0,1]^N,$$

where $\lambda, \gamma$ are the two pacing parameters for sampling based on *easiness* and the *true diverseness* as sample importance priors, respectively. The negative $l_1$-norm: $-\|\mathcal{W}\|_1$ is used to select easy samples over hard samples, as seen in conventional SPL. The negative $l_2$-norm inherited from the original SPLD algorithm is used to disperse non-zero elements of $\mathcal{W}$ across a large number of clusters to obtain a diverse selection of training samples. The student CNN receives a diverse cluster of samples, the up-to-date model parameters $\theta$, $\lambda$, $\gamma$ and outputs the optimal of $\mathcal{W}$ of $\min_{\mathcal{W}} \mathbb{E}(\theta, \mathcal{W}; \lambda, \gamma)$ for extracting the global optimum of this optimization problem. The detailed algorithm to train the student CNN with LEAP is presented in Algorithm 2.

---

**Algorithm 2:** Train Student CNN with LEAP.

**Input** : $\mathcal{D}$ (training data), $B$ (mini-batch size), $E$ (epochs), $\{\beta_1, \beta_2\}$ (self-pace parameters)
**Output:** The model parameters $\theta$.

1  Initialize the student CNN model.
2  Initialize $N$. /* Total training samples in $\mathcal{D}$ */
3  Initialize $\mathcal{W}^*, \lambda, \gamma$ ; /* Assign initial values of the pace parameters. */
4  Get the mini-batches sequence $\mathcal{D} = \left\{ \{x_B\}_1, \ldots, \{x_B\}_{\frac{N}{B}} \right\}$.
5  /* Train Embedding CNN asynchronously in parallel with the
6  Student CNN using multiprocessing. */
7  Start constructing the representation space using **Algorithm 1**.
8  **for** *each epoch* $\{e = 1, \ldots, E\}$ **do**
9  $\quad$ **for** *each mini-batch* $\left\{ \{x_B\}_{\frac{N}{B}} = 1, \ldots, \frac{N}{B} \right\}$ **do**
10 $\quad\quad$ $\theta^* = \underset{\theta}{\arg\min}\, \mathbb{E}(\theta, \mathcal{W}^*; \lambda, \gamma)$ /* Train Student CNN. */
11 $\quad\quad$ /* Solve $\min_{\mathcal{W}} \mathbb{E}(\theta, \mathcal{W}; \lambda, \gamma)$. */
12 $\quad\quad$ Get clusters $\{D^k\}_{k=1}^K$ from the embedding CNN.
13 $\quad\quad$ **for** *each cluster* $\{k = 1, \ldots, K\}$ **do**
14 $\quad\quad\quad$ Sort training samples in $D^k$ in increasing order of their loss values, $\mathcal{L}$.
15 $\quad\quad\quad$ Represent the labels of $D^k$ as $\{y_1^k, \ldots, y_{n_k}^k\}$ and weights as $\{\mathcal{W}_1^k, \ldots, \mathcal{W}_{n_k}^k\}$.
16 $\quad\quad\quad$ **for** *each sample* $i = \{1, \ldots, n_k\}$ **do**
17 $\quad\quad\quad\quad$ /* Selects the sample based on easiness and true
$\quad\quad\quad\quad\quad$ diverseness. */
18 $\quad\quad\quad\quad$ **if** $\mathcal{L}(y_i^k, f(x_i^k, \theta)) < (\lambda + \gamma \frac{1}{\sqrt{i}+\sqrt{i-1}})$ **then** $\mathcal{W}_i^k = 1$ ;
19 $\quad\quad\quad\quad$ **else** $\mathcal{W}_i^k = 0$ ;
20 $\quad\quad\quad$ **end**
21 $\quad\quad$ **end**
22 $\quad\quad$ $\mathcal{W}^* = \min_{\mathcal{W}} \mathbb{E}(\theta, \mathcal{W}; \lambda, \gamma)$ ;
23 $\quad\quad$ $\lambda \leftarrow \beta_1 \lambda ; \gamma \leftarrow \beta_2 \lambda$ ; /* Update the learning pace for $\lambda$ and $\gamma$. */
24 $\quad$ **end**
25 $\quad$ $\theta = \theta^*$
26 $\quad$ $N = \{x_n\}_{n=1}^N * \mathcal{W}^*$ /* Training samples for next epoch $e$. */
27 **end**

---

For any $\theta$, we show how we can solve for the global optimum to $\min_{\mathcal{W}} \mathbb{E}(\theta, \mathcal{W})$ in linearithmic time. Step 18 of Algorithm 2 selects the "easy" samples for training when $\mathcal{L}(y_i^k, f(x_i^k, \theta)) < \lambda$, thus $\mathcal{W}_i = 1$. Otherwise in Step 19, $\mathcal{W}_i = 0$ if the $\mathcal{L}(y_i^k, f(x_i^k, \theta)) < \lambda + \gamma$, which represents the "hard" samples with higher losses. We select other samples by ranking a sample w.r.t to its loss value within its cluster, denoted by $i$. Then, we compare the losses to a threshold $\lambda + \gamma \frac{1}{\sqrt{i} + \sqrt{i-1}}$. Step 18 penalizes samples repeatedly selected from the same cluster, seeing as this threshold decreases as the sample's rank $i$ grows.

## 4 EXPERIMENTS

All experiments were conducted using the PyTorch framework, while leveraging containerized multi-GPU training on NVIDIA P100 Pascal GPUs through Docker. We compared our LEAP framework against the original SPLD algorithm and Random sampling on MNIST, FashionMNIST and CIFAR-10. The $\lambda$ and $\gamma$ pace parameters are kept consistent between the SPLD strategy in LEAP and the original SPLD algorithm to ensure a fair evaluation. The embedding CNN is trained asynchronously in parallel with the student CNN. The computational requirement of training the embedding CNN can be mitigated by leveraging multiprocessing for parallel computing to share data between processes locally using arrays and values. As a result, at every epoch, the student CNN adaptively selects "easy" samples from $K$ cluster representations generated by the embedding CNN. In our experiments, we mainly compare convergence in terms of number of mini-batches required to achieve a comparable or better state-of-the-art test performance. We also visualize the original high-dimensional representations using t-SNE (van der Maaten & Hinton, 2008), where the different colours correspond to different classes and the values to density estimates. The following sections discuss the experimental setups and results in more detail.

### 4.1 LEAP WITH LENET + LENET ON MNIST

In the experiments for MNIST, we extract the feature embeddings from a LeNet (LeCun et al., 1998), as the embedding CNN, and learn a representation space using the Magnet Loss. The fully-connected layer of the LeNet is replaced with an embedding layer for compacting the distribution of the learned features for feature similarity comparison using the Magnet Loss. The student CNN (classifier) was also a LeNet which we then trained with our LEAP framework. The embedding CNN was trained with randomly sampled mini-batches of size 64 and optimized using Adam with a learning rate of 0.0001. The results in Figure 2 show that the test performance is comparable to that of Random sampling and shows better convergence than the standard SPLD algorithm. The learned representation space of the MNIST dataset using LeNet is presented in Figure 3b and the training loss in Figure 3a. The MNIST experiments were primarily carried out to show that the LEAP framework can be deployed as an end-to-end DNN training protocol.

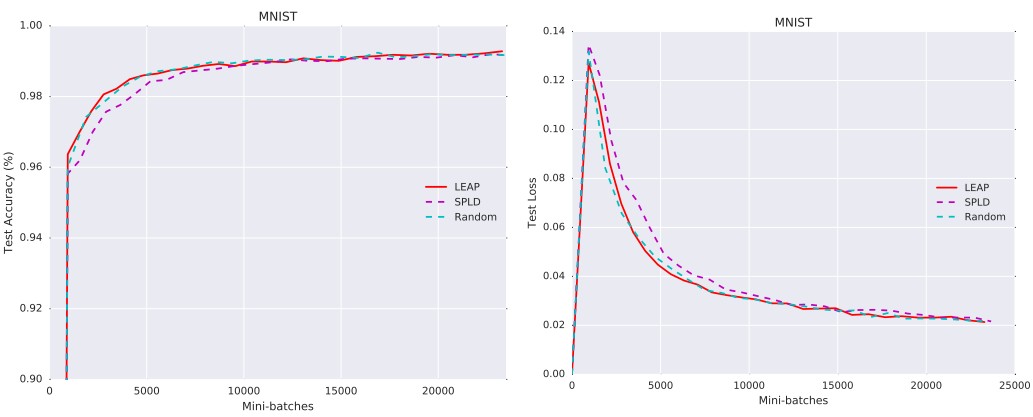

FIGURE 2: Averaged results of 5 independent runs on MNIST. The solid line represents our LEAP framework.

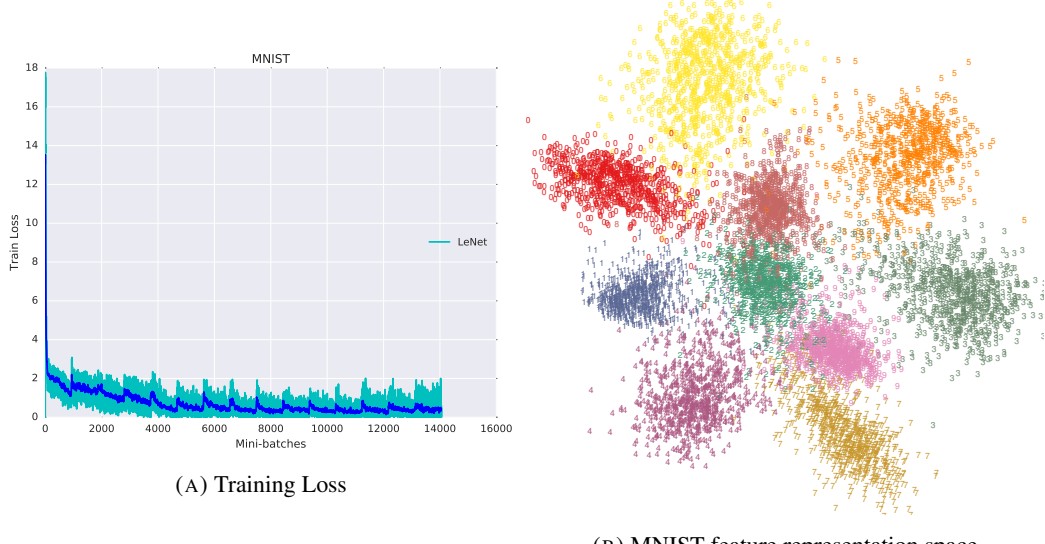

(A) Training Loss

(B) MNIST feature representation space

FIGURE 3: (a) The training loss with the moving average and (b) the feature representation space learned by LeNet (embedding CNN) using the LEAP framework on MNIST.

## 4.2 LEAP WITH RESNET-18 + LENET ON FASHIONMNIST

The experiments for FashionMNIST had a similar setup to MNIST, however we use a ResNet-18 (He et al., 2016) for the classifier. The embedding CNN remains the same LeNet for feature extraction, and is trained using an identical setup to the MNIST experiments. The classifier is trained using SGD with Nesterov momentum of 0.9, weight decay of 0.0005 and a learning rate of 0.001. The FashionMNIST dataset is considered a direct drop-in replacement for the original MNIST dataset, with a training set of 60,000 examples and a test set of 10,000 examples. Each example is a 28×28 grayscale image, associated with a label from 10 classes. We performed data augmentation on the training set with normalization, random horizontal flip, random vertical flip, random translation, random crop and random rotation. In comparison to SPLD and Random sampling, the results in Figure 4 reveal that LEAP converges to a higher test accuracy with a fewer number of mini-batch updates before saturating. Figure 5b depicts a learned representation space of the FashionMNIST dataset and the training loss of the embedding CNN in Figure 5a.

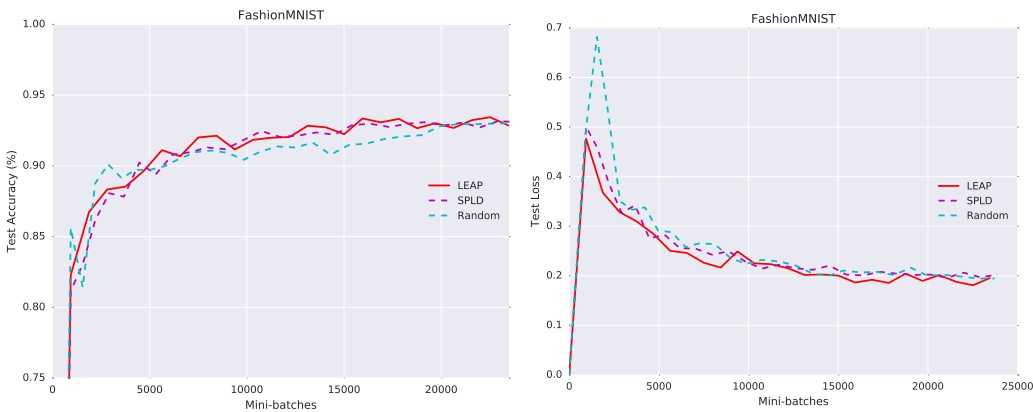

FIGURE 4: Averaged results of 5 independent runs on FashionMNIST. The solid line represents our LEAP framework.

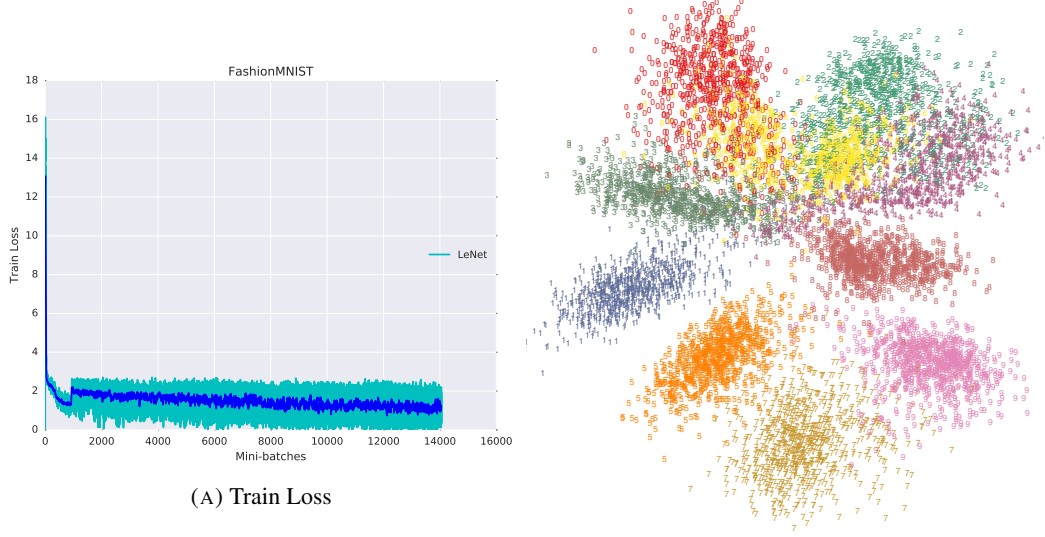

(A) Train Loss

(B) FashionMNIST feature representation space

FIGURE 5: (a) The train loss with the moving average and (b) the feature representation space learned by LeNet (embedding CNN) using the LEAP framework on FashionMNIST.

## 4.3 LEAP WITH VGG-16 + RESNET-18 ON CIFAR-10

We ran two sets of experiments on CIFAR-10, one with a fixed learning rate and the other with a learning rate scheduler identical to that of the WideResNet (Zagoruyko & Komodakis, 2016) training scheme. The CIFAR-10 training set was augmented with normalization, random horizontal flip and random crop. We used VGG-16 as our embedding CNN and ResNet-18 as our classifier in the CIFAR-10 experiments. We chose ResNet-18 over other architectures because it is faster to train and achieves good performance on CIFAR-10. Our experiments revealed that VGG-16 (Simonyan & Zisserman, 2014) learned strong and rich feature representations which yielded the best convergence on the Magnet Loss. Therefore, we treat the VGG-16 model as a feature extraction engine and use it for feature extraction from CIFAR-10 images, without any fine-tuning. In the first experiment, the classifier is trained using SGD with a momentum of 0.9, weight decay of 0.0005, a fixed learning rate of 0.001 and batch size of 128. In the second experiment, the classifier is trained with batch sizes of 128 as well, using SGD with a momentum of 0.9, weight decay of 0.0005 and a starting learning rate of 0.1 which is dropped by a factor of 0.1 at 60, 120 and 160 epochs. In our experiments, this would translate to 23,400, 46,800, and 62,400 mini-batch updates. In both experiments, VGG-16 is trained with randomly sampled mini-batches of size 64 and optimized using Adam at a learning rate of 0.0001.

At a fixed learning rate of 0.001, training ResNet-18 with the LEAP framework results in a faster convergence to achieve higher test accuracy than either Random sampling or SPLD (Figure 6). Interestingly, SPLD and Random sampling show comparable results on an average of 5 runs. This is because SPLD uses $K$-Means to partition its data into $K$ clusters at the start of training, which would lead to sub-optimal clustering results containing samples that are not of similar-instance level. As a result, the SPLD would not be selecting diverse samples of similar-instance level when traversing through the different clusters. On the other hand, our LEAP framework ensures that optimal clustering is achieved using the Magnet Loss and the classifier is able to use the learned representation space adaptively as training progresses. This ensures that during early stages of training, the mini-batches being fed into the student CNN are parameterized with truly diverse and easy samples. As the model matures, the mini-batches maintain diversity, as well as a mix of easy and hard samples.

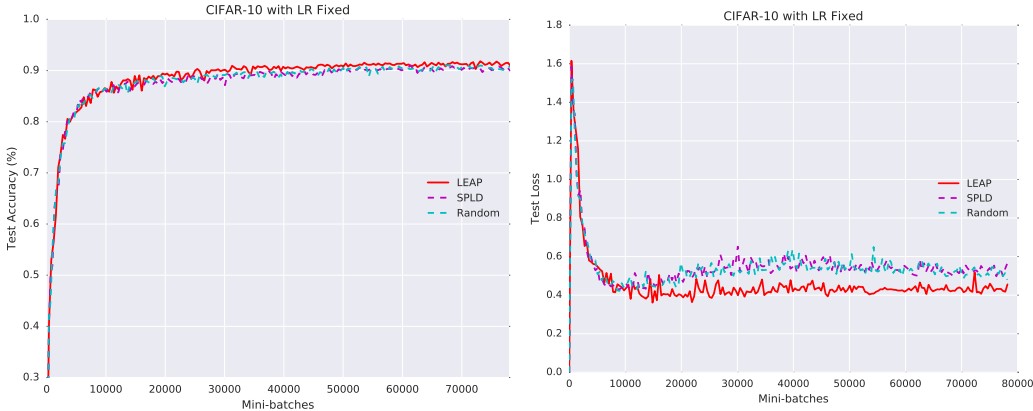

FIGURE 6: Averaged results of 5 independent runs on CIFAR-10 with a fixed learning rate of 0.001. The solid line represents our LEAP framework.

The second set of experiments we ran on CIFAR-10 with the learning rate scheduler shows that LEAP converges faster earlier on in training compared to SPLD and Random sampling (Figure 7). As the learning rate drops by a factor of 0.1 at 23,400, 46,800, and 62,400 mini-batch updates, the classifier under the LEAP training protocol eventually achieves a higher test accuracy than SPLD and Random sampling. Also, the test loss of LEAP is lower than that of SPLD and Random. This is expected because LEAP is designed to sample for heterogeneity, thus maximizing the diversity relevant to the learning stage of the student CNN, which also helps prevent overfitting. The learned feature representation space of the augmented CIFAR-10 data is shown in Figure 8b and the training loss of the embedding CNN is presented in Figure 8a.

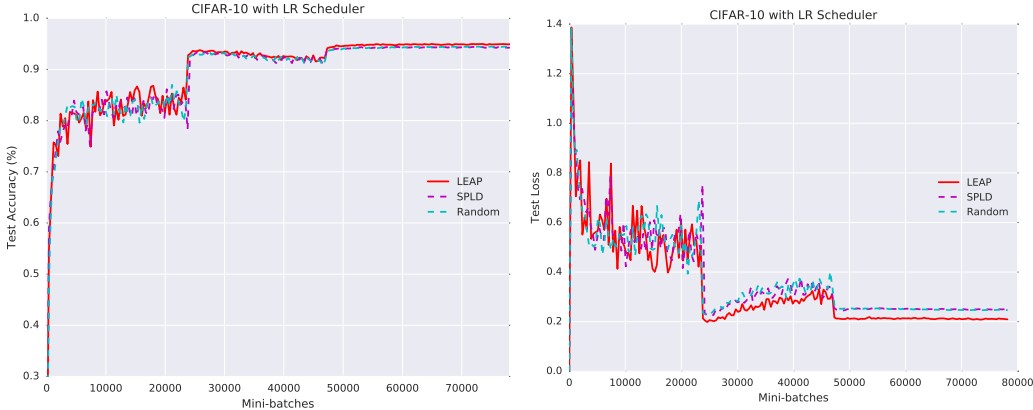

FIGURE 7: Averaged results of 5 independent runs on CIFAR-10 with a learning rate scheduler identical to the WideResNet training scheme. The solid line represents our LEAP framework.

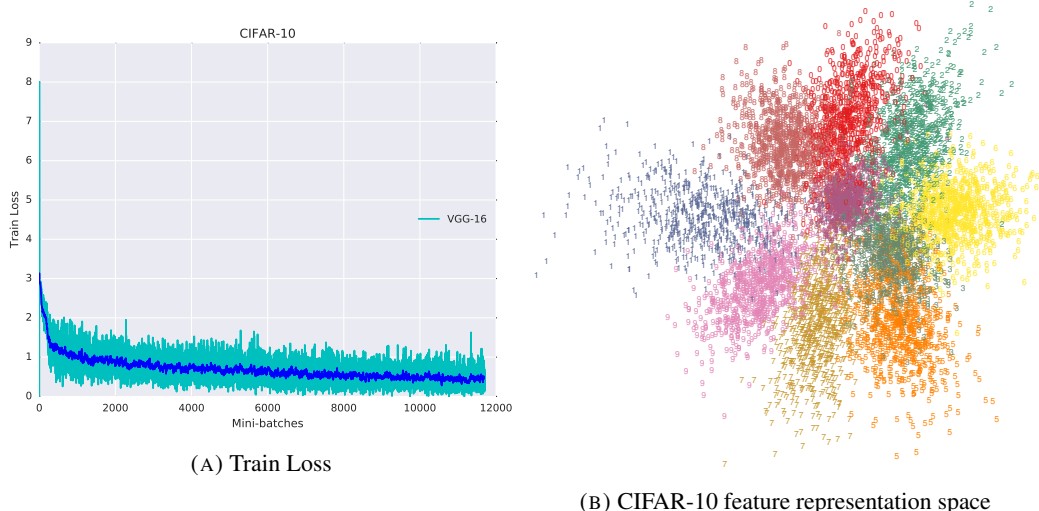

(A) Train Loss

(B) CIFAR-10 feature representation space

FIGURE 8: (a) The train loss with the moving average and (b) the feature representation space learned by VGG-16 (embedding CNN) using the LEAP framework on CIFAR-10.

## 4.4 LEAP WITH VGG-16 + WIDERESNET-28-10 ON CIFAR-100

We evaluated our LEAP framework on CIFAR-100 dataset with a WideResNet (student CNN) that has a fixed depth of 28, a fixed widening factor of 10, and dropout probability of 0.3. The embedding CNN was a VGG-16 setup similar to our CIFAR-10 experiments. The optimizer and learning rate scheduler used for training the WideResNet was identical to the CIFAR-10 experiments. A data augmentation scheme was not applied on the CIFAR-100 dataset. The results in Figure 9 reveal that on CIFAR-100, a dataset that requires a lot more fine-grained recognition, there is noticeable gain in improvement over Random and SPLD. We can expect this gain on a more fine-grained dataset because the Magnet loss is a metric embedding technique which performs really well on classification tasks for fine-grained visual recognition (Rippel et al., 2015). This further shows that the combination of a dynamic representation space and self-paced strategy can lead to a noticeable improvement on more complex fine-grained datasets.

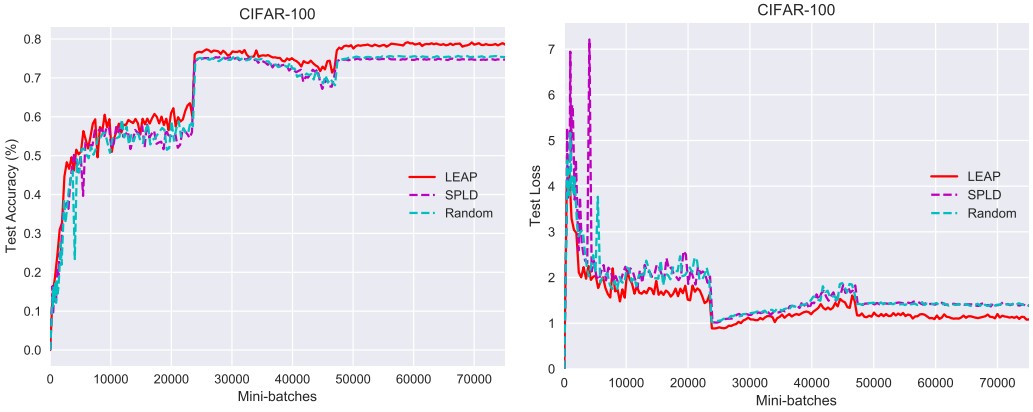

FIGURE 9: Averaged results of 4 independent runs on CIFAR-100 with a learning rate scheduler identical to the WideResNet training scheme. The solid line represents our LEAP framework.

## 4.5 LEAP WITH VGG-16 + WIDERESNET-16-8 ON SVHN

The Street View House Numbers (SVHN) dataset (Netzer et al., 2011) is a real-world image dataset with 630,420 RGB images of $32 \times 32$ pixels in size, where each image consists of digits that are from one of ten different classes. A single image may contain multiple digits, and the task is to classify the digit located at the center of the image. The SVHN dataset is split into the training set, testing set and extra set with 73,257, 26,032, and 531,131 images, respectively. The student CNN used to train on the SVHN dataset was a WideResNet with a fixed depth of 16, a fixed widening factor of 8, and dropout probability of 0.4. The SVHN training set and extra set were combined for a total of 604,388 images to train the student CNN for 65 epochs and no data augmentation scheme was applied. The student model was optimized using SGD with Nesterov momentum of 0.9, weight decay of 0.0005 and batch-size of 128. The embedding CNN was a VGG-16 setup similar to our CIFAR-10 and CIFAR-100 experiments. The results in Figure 10 show that LEAP is able to converge faster to a higher test accuracy than Random and SPLD. The learned feature representation space of the SVHN data is presented in Figure 11b and the training loss of the embedding CNN is presented in Figure 11a.

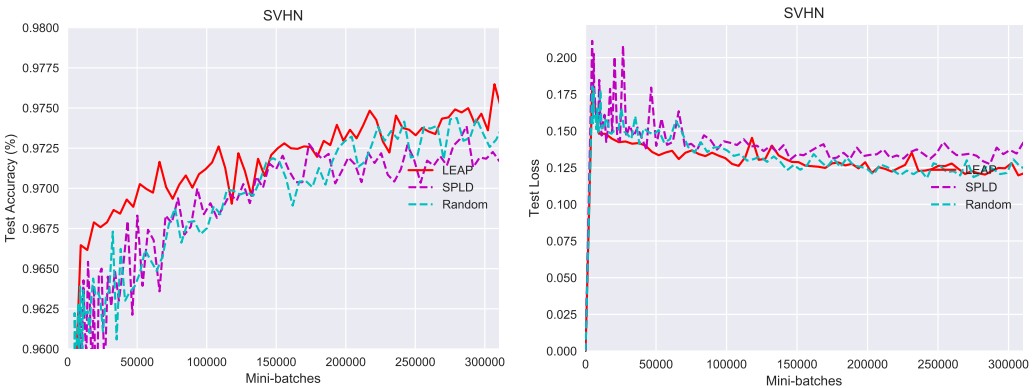

FIGURE 10: Averaged results of 4 independent runs on SVHN. The solid line represents our LEAP framework.

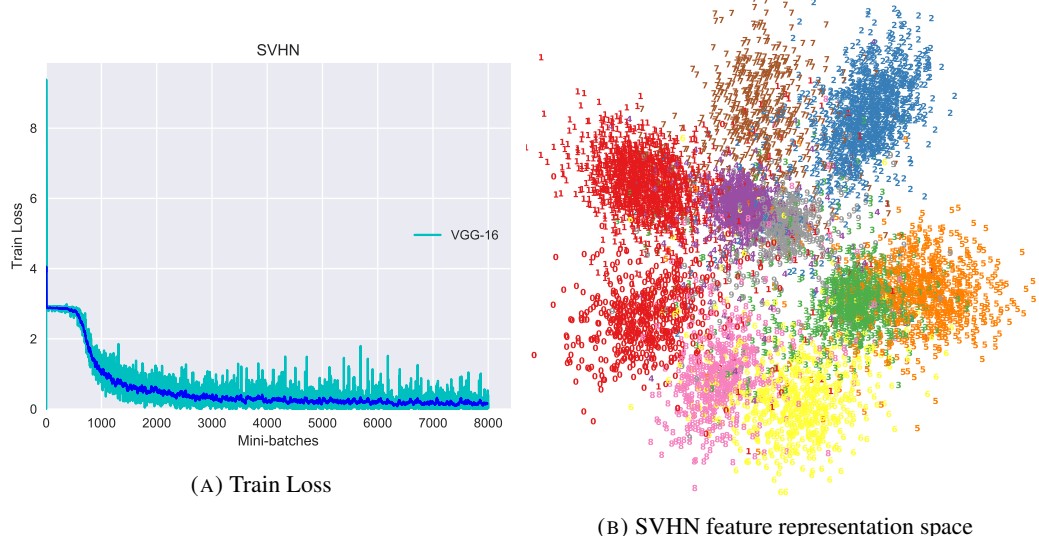

(A) Train Loss

(B) SVHN feature representation space

FIGURE 11: (a) The train loss with the moving average and (b) the feature representation space learned by VGG-16 (embedding CNN) using the LEAP framework on SVHN.

## 5 DISCUSSION AND FUTURE WORK

An important finding is that fusing a salient non-linear representation space with a dynamic learning strategy can help a DNN converge towards an optimal solution. A random curriculum or a dynamic learning strategy without a good representation space was found to achieve a lower test accuracy or converge more slowly than LEAP. Biasing samples based on the *easiness* and *true diverseness* to select mini-batches shows improvement in convergence to achieve classification performance comparable or better than the baselines, Random and SPLD. As shown in Table 1, the student CNN models show increased accuracy on MNIST, Fashion-MNIST, CIFAR-10, CIFAR-100 and SVHN with our LEAP sampling method. It is to be noted that the LEAP framework improves the performance of complex convolutional architectures which already leverage regularization techniques such as batch normalization, dropout, and data augmentation. We see that the improvements on coarse-grain datasets such as MNIST, Fashion-MNIST, CIFAR-10, and SVHN are between 0.11 and 0.81 percentage points. On a fine-grained dataset like CIFAR-100, it is more challenging to obtain a high classification accuracy. This is because there are a 100 fine-grained classes but the number of training instances for each class is small. We have only 500 training images and 100 testing images per class. In addition, the dataset contains images of low quality and images where only part of the object is visible (i.e. for a person, only head or only body). However, we show that with LEAP, we can attain a significant increase in accuracy by 4.50 and 3.72 percentage points over the baselines SPLD and Random, respectively. The mix of easy and diverse samples from a more accurate representation space of the data helps select appropriate samples during different stages of training and guide the network to achieve a higher classification accuracy, especially for more difficult fine-grained classifcation tasks.

| Sampling Method | | MNIST | Fashion-MNIST | CIFAR-10 | CIFAR-100* | SVHN* |
|---|---|---|---|---|---|---|
| **LEAP** | | **99.31 $\pm$ 0.06** | **93.76 $\pm$ 0.11** | **95.12 $\pm$ 0.16** | **79.17 $\pm$ 0.24** | **97.65 $\pm$ 0.09** |
| SPLD | | 99.20 $\pm$ 0.05 | 93.17 $\pm$ 0.18 | 94.42 $\pm$ 0.21 | 74.67 $\pm$ 0.30 | 97.38 $\pm$ 0.06 |
| Random | | 99.24 $\pm$ 0.06 | 92.95 $\pm$ 0.14 | 94.49 $\pm$ 0.12 | 75.45 $\pm$ 0.25 | 97.43 $\pm$ 0.11 |

TABLE 1: Experimental results across all datasets (MNIST, Fashion-MNIST, CIFAR-10, CIFAR-100, and SVHN) and sampling methods (LEAP, SPLD, and Random). The test accuracy (%) results are averaged over five runs, with the exception of CIFAR-100 and SVHN which had four runs each. "*" indicates that no data augmentation scheme was applied on the dataset.

In cases where the classification dataset is balanced and the classes are clearly identifiable, we showed that our end-to-end LEAP training protocol is practical. An interesting line of work would be to apply LEAP on more complex real-world classification datasets such as iNaturalist (Horn et al., 2017), where there are imbalanced classes with a lot of diversity and require fine-grained visual recognition. Another interesting area of application would be learning representations using DML for different computer vision tasks (e.g. human pose estimation, human activity recognition, semantic segmentation, etc.) and fusing a representative SPL strategy to train the student CNN.

## 6 CONCLUSION

We introduced LEAP, an end-to-end representation learning SPL strategy for adaptive mini-batch formation. Our method uses an embedding CNN for learning an expressive representation space through a DML technique called the Magnet Loss. The student CNN is a classifier which can exploit this new knowledge from the representation space to place the *true diverseness* and *easiness* as sample importance priors during online mini-batch selection. The computational overhead of training two CNNs can be mitigated by training the embedding CNN and student CNN in parallel. LEAP achieves good convergence speed and higher test performance on MNIST, FashionMNIST, CIFAR-10, CIFAR-100 and SVHN using a combination of two deep CNN architectures. We hope this will help foster progress of end-to-end SPL fused DML strategies for DNN training, where a number of potentially interesting directions can be considered for further exploration. Our framework is implemented in PyTorch and will be released as open-source on GitHub following the review process.

ACKNOWLEDGMENTS

Acknowledgements were removed for double blind review.

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
