# OpenReview forum: "LEAP: Learning Embeddings for Adaptive Pace"
_ICLR.cc/2018/Conference — Reject_

### Official Review · AnonReviewer1 · 2017-11-27

**Rating:** 3
**Confidence:** 4

**Review:**

(Summary)
This paper is about learning a representation with curriculum learning style minibatch selection in an end-to-end framework. The authors experiment the classification accuracy on MNIST, FashionMNIST, and CIFAR-10 datasets.

(Pros)
The references to the deep metric learning methods seem up to date and nicely summarizes the recent literatures.

(Cons)
1. The method lacks algorithmic novelty and the exposition of the method severely inhibits the reader from understand the proposed idea. Essentially, the method is described in section 3. First of all, it's not clear what the actual loss the authors are trying to minimize. Also, \min_v E(\theta, v; \lambda, \gamma) is incorrect. It looks to me like it should be E \ell (...) where \ell is the loss function.

2. The experiments show almost no discernable practical gains over 'random' baseline which is the baseline for random minibatch selection.

(Assessment)
Clear rejection. The method is poorly written, severely lacks algorithmic novelty, and the proposed approach shows no empirical gains over random mini batch sampling.

---

### Official Review · AnonReviewer2 · 2017-12-01
**The authors propose a method that uses an embedding network trained with magnet loss for adaptively sampling and feeding the student network that is being trained for the actual task**

**Rating:** 4
**Confidence:** 4

**Review:**

While the idea is novel and I do agree that I have not seen other works along these lines there are a few things that are missing and hinder this paper significantly.

1. There are no quantitative numbers in terms of accuracy improvements, overhead in computation in having two networks.
2. The experiments are still at the toy level, the authors can tackle more challenging datasets where sampling goes from easy to hard examples like birdsnap. MNIST, FashionMNIST and CIFAR-10 are all small datasets where the true utility of sampling is not realized. Authors should be motivated to run the large scale experiments.

---

### Official Review · AnonReviewer4 · 2017-12-03

**Rating:** 6
**Confidence:** 3

**Review:**

The authors purpose a method for creating mini batches for a student network by using a second learned representation space to dynamically selecting  examples by their 'easiness and true diverseness'. The framework is detailed and results on MNIST, cifar10 and fashion-MNIST are presented. The work presented is novel but there are some notable omissions:
 - there are no specific numbers presented to back up the improvement claims; graphs are presented but not specific numeric results
- there is limited discussion of the computational cost of the framework presented
- there is no comparison to a baseline in which the additional learning cycles used for learning the embedding are used for training the student model.
- only small data sets are evaluated. This is unfortunate because if there are to be large gains from this approach, it seems that they are more likely to be found in the domain of large scale problems, than toy data sets like mnist.

**edit
In light of the changes made, and in particular the performance gains achieved on CIFAR-100, i have increased my ratting from a 4 to a 6

---

### Author Response · Authors · 2018-01-05
**Response to all reviewers (Part 2)**



R1, R2, R4 Quantitative results to backup improvement claims

A table with a summary of the experimental results is provided in Section 5. Please refer to the latest revision for the updated Table 1. Here, we present the test accuracy (%) results across all datasets including: MNIST, Fashion-MNIST, CIFAR-10, CIFAR-100, and SVHN for the following sampling methods: Learning Embeddings for Adaptive Pace (LEAP), Self-Paced Learning with Diversity (SPLD), and Random. The test accuracy results of MNIST, Fashion-MNIST, and CIFAR-10 are averaged over 5 runs. The results for CIFAR-100 and SVHN are averaged over 4 runs. The results show that there is a noticeable increase in test performance across all datasets with the LEAP dynamic sampling strategy, especially for the CIFAR-100 dataset.

R2, R4 Computational cost of this framework

We agree that training two complex CNN architectures (i.e. VGG-16, ResNet-18, etc.) would raise concerns for overhead in computation. However, we would like to clarify that the embedding CNN and student CNN are asynchronously trained in parallel by using multiprocessing to share data between processes in a local environment using arrays and values. The idea is to have an embedding CNN that is adaptively sculpting a representation space, while the student CNN is being trained. The student CNN leverages the $K$ cluster representations constructed by the embedding CNN, to select samples based on the “easiness” from each of the $K$ clusters for each class, $c$ in $C$ classes. This way we are ensuring that the samples that the student model considers “easy” also maintains diversity, which is important for constructing mini-batches iteratively. Therefore, the extra training cost of the embedding CNN can be mitigated by having it train in parallel to the actual classification model. This setup is more apparent in Section 3, which contains more specific and updated details of the methodology for both the embedding CNN and student CNN.

R1, R2, R4 Experiments on complex datasets

We conducted experiments on two additional datasets, SVHN and CIFAR-100 which is considered a more fine-grained visual recognition dataset. We used a WideResNet for the student CNN and VGG-16 for the embedding CNN to train on CIFAR-100 using LEAP. The specific training scheme used for CIFAR-100 is detailed in Section 4.4.  The CIFAR-100 experiments revealed that we achieve a noticeable gain in performance when using the LEAP framework with a test accuracy of 79.17% \pm 0.24%. The LEAP framework outperforms the baselines, SPLD and Random, by 4.50% and 3.72%, respectively. Effectively, we saw that on a more challenging fine-grained classification task, the LEAP framework performs really well. While we agree with the reviewers that the true utility of our framework can be realized in large-scale problems (i.e. BirdSnap, ImageNet, etc.), we have yet to perform those experiments.

The MNIST experiments were mainly performed to show that the LEAP framework can be employed end-to-end for a simple supervised classification task. Then, we extended this to Fashion-MNIST which is considered a direct drop-in replacement for MNIST. Fashion-MNIST served to be another small classification dataset that can be used to test and verify the feasibility of our approach, which also served to be successful. CIFAR-10 experiments showed that we can learn a representation space with $K$ clusters for each class in the dataset, by extracting features from RGB images and computing the Magnet loss with the embedding CNN. Then, we showed that we can use this learned representation space to adaptively sample “easy” training instances diversely from $K$ clusters for each classified class.

---

### Author Response · Authors · 2018-01-05
**Response to all reviewers (Part 1)**

We thank all of the reviewers for their careful review of our paper, and for the valuable comments and constructive criticism that ensued. We performed a major revision to the paper to take all of them into account, and in the process, we believe the paper has improved significantly. These are detailed below:

R1 Methodology clarification

We made significant updates to the methodology in Section 3. In Section 3.1, we provide a detailed training algorithm for the embedding CNN which uses the Magnet loss to form a representation space consisting of $K$ clusters for $C$ classes by adaptive density discrimination. This results in a training set $D$ partitioned into learned representation space, $D_K^c$, while maintaining 	intra-class variation and inter-class similarity. The details of the objective function for the LEAP framework are added in Section 3.2, which is given by:

\min_{\theta, \mathcal{W}} \mathbb{E}(\theta, \mathcal{W}; \lambda, \gamma) = \sum_{i=1}^{n}w_i\mathcal{L}(y_i, f(x_i,\theta)) - \lambda \sum_{i=1}^{n}w_i - \gamma\|\mathcal{W}\|_{2,1}, \ \text{s.t} \ \mathcal{W} \in [0,1]^{n}

In LEAP, we assume that a dataset contain $N$ samples, $\mathcal{D} = \{\mathbf{x}_n\}_{n=1}^{N}$, is grouped into $K$ clusters for each class $c$ through the Magnet loss to get: $\{\mathcal{D}^{k}\}_{k=1}^K$, where $\mathcal{D}^{k}$ corresponds to the $k^{th}$ cluster, $n_k$ is the number of samples in each cluster and $\sum_{k=1}^{K}n_k = N$. A weight vector is  $\mathcal{W}^{k} = (\mathcal{W}_1^k,\ldots,\mathcal{W}_{n_k}^k)^T$, where each $\mathcal{W}_{n_k}^k$ is assigned a weight $[0,1]^{n_k}$ for each sample in cluster $k$ for $K$ clusters.

The easiness and true diverseness terms are given by $\lambda$ and $\gamma$.  We use the negative $l_1$-norm: $-\|\mathcal{W}\|_1$ to select easy samples over hard samples. The negative $l_2$-norm is used to disperse non-zero elements of the weights $\mathcal{W}$ across a large number of clusters so that we can get a diverse set of training samples.

In addition, we give specific details on the LEAP algorithm (Section 3.2) for training the student CNN, where we indicate how the embedding CNN and student CNN are used in conjunction. In this subsection, we also present the self-paced sample selection strategy, which specifies how the training samples are selected based on the “easiness” and “true diverseness” according to the student CNN model, such that we solve $\min_{\mathcal{W}}\mathbb{E}(\theta, \mathcal{W}; \lambda, \gamma)$. If the cross-entropy loss, $\mathcal{L}(y_i^{k}, f(x_i^{k},\theta))$, is less than $(\lambda + \gamma\frac{1}{\sqrt{i}+\sqrt{i-1}})$, then we assign a weight $\mathcal{W}_i^{k} = 1$, otherwise $\mathcal{W}_i^{k} = 0$. $i$ is the training instance’s rank w.r.t. its cross-entropy loss value within its cluster. The instance with a smaller loss than the assigned threshold will be selected during training. Therefore, the new $\mathcal{W}$ becomes equal to  $\min_{\mathcal{W}}\mathbb{E}(\theta, \mathcal{W}; \lambda, \gamma)$. Next, we update the learning pace for $\lambda$ and $\gamma$.

---

### Decision · Program_Chairs · 2018-01-29
**ICLR 2018 Conference Acceptance Decision**

**Decision:**

Reject

**Comment:**

Although paper has been improved with new quantitative results and additional clarity, the reviewers agree though that larger-scale experiments would better highlight the utility of the method. There are some concerns with computational cost, despite the fact that the two networks are trained asynchronously. A baseline against a single, asynchronously trained network (multiple GPUs) would help strengthen this point. Some reviewers expressed concerns with novelty.